A new lineage of Cretaceous jewel wasps (Chalcidoidea: Diversinitidae)

Haas Michael michael.haas@smns-bw.de 1
Burks Roger A. 2
Krogmann Lars lars.krogmann@smns-bw.de 1
1 Department of Entomology, State Museum of Natural History Stuttgart , Stuttgart , Germany
2 Department of Entomology, University of California, Riverside , Riverside , United States of America
De Baets Kenneth
Electronic publication date: 2018 Apr 16
Publication date: 2018
Volume: 6
Electronic Location ID: e4633
Received 2017 Nov 30; Accepted 2018 Mar 28
Copyright: ©2018 Haas et al.
Copyright year: 2018
Copyright holder: Haas et al.
License: This is an open access article distributed under the terms of the Creative Commons Attribution License, which permits unrestricted use, distribution, reproduction and adaptation in any medium and for any purpose provided that it is properly attributed. For attribution, the original author(s), title, publication source (PeerJ) and either DOI or URL of the article must be cited.
License URL: https://creativecommons.org/licenses/by/4.0/

Keywords: Burmese amber, Hymenoptera, Systematic Paleontology, Glabiala, Burminata, Diversinitus, New genera, Ground plan biology, Parasitoids

Funding: NSF DEB 1555808 Funding for Roger A. Burks was provided by NSF DEB 1555808. The funders had no role in study design, data collection and analysis, decision to publish, or preparation of the manuscript.

==============================
Jewel wasps (Hymenoptera: Chalcidoidea) are extremely species-rich today, but have a sparse fossil record from the Cretaceous, the period of their early diversification. Three genera and three species, Diversinitus attenboroughi gen. & sp. n., Burminata caputaeria gen. & sp. n. and Glabiala barbata gen. & sp. n. are described in the family Diversinitidae fam. n., from Lower Cretaceous Burmese amber. Placement in Chalcidoidea is supported by the presence of multiporous plate sensilla on the antennal flagellum and a laterally exposed prepectus. The new taxa can be excluded from all extant family level chalcidoid lineages by the presence of multiporous plate sensilla on the first flagellomere in both sexes and lack of any synapomorphies. Accordingly, a new family is proposed for the fossils and its probable phylogenetic position within Chalcidoidea is discussed. Morphological cladistic analyses of the new fossils within the Heraty et al. (2013) dataset did not resolve the phylogenetic placement of Diversinitidae, but indicated its monophyly. Phylogenetically relevant morphological characters of the new fossils are discussed with reference to Cretaceous and extant chalcidoid taxa. Along with mymarid fossils and a few species of uncertain phylogenetic placement, the newly described members of Diversinitidae are among the earliest known chalcidoids and advance our knowledge of their Cretaceous diversity.

Introduction

Jewel wasps (Hymenoptera: Chalcidoidea) are estimated to constitute one of the most species-rich insect lineages. Estimated numbers range from 100,000 to 500,000 species, which may comprise 10% of insect diversity, though only about 22,000 species have been described to date (Noyes, 1978; Noyes, 2000; Noyes, 2017; Heraty & Gates, 2001). Their evolutionary success is mirrored by and likely results from their varied biological life styles. Jewel wasps develop mainly as parasitoids of 13 different insect orders, as well as some nematodes, pseudoscorpions and arachnids, and thus are essential beneficial regulators, while some species are phytophages or even obligate pollinators of figs (Ficus) (Gibson, Heraty & Woolley, 1999; Weiblen, 2002; Heraty, 2009). Despite recent progress (Munro et al., 2011; Heraty et al., 2013; Peters et al., 2018), the relationships among most chalcidoid taxa as well as their evolutionary history still remain unresolved. The role of fossils in a phylogenetic framework is pivotal in understanding some of the evolutionary processes that led to chalcidoid megadiversity and provide valuable information on morphological character evolution (Donoghue et al., 1989; Peters et al., 2018). Reliably placed fossils can shed light on the minimum age of taxa and allow calibrations of molecular phylogenies to resolve timing and patterns of biological shifts (Ware & Barden, 2016; Gunkel et al., 2017; Slater, Harmon & Alfaro, 2012).

Numerous chalcidoid fossils have been reported from different amber deposits (Grimaldi & Engel, 2005; Penney, 2010), but few of them have been formally described. Most described chalcidoids stem from young (Eocene and Miocene) deposits, which already host an astonishing phylogenetic diversity of taxa (Darling, 1996; Gibson, 2008; Gibson, 2009; Gibson, 2013; Engel, 2009; Engel, McKellar & Huber, 2013; Heraty & Darling, 2009; Compton et al., 2010; McKellar & Engel, 2012; Krogmann, 2013; Simutnik, Perkovsky & Gumovsky, 2014; Bläser, Krogmann & Peters, 2015; Burks et al., 2015; Farache et al., 2016). It is believed that most chalcidoid families diversified after the Upper Cretaceous (Heraty et al., 2013; Peters et al., 2018) during a period that falls within a major gap in the chalcidoid fossil record, from which only few taxa have been described or even discovered (Burks et al., 2015; Heraty & Darling, 2009; Penney, 2010). The earliest reported and described chalcidoids date back to the Lower Cretaceous period, 106–115 million years ago (mya) (Kaddumi, 2005; Grimaldi & Engel, 2005; Penney, 2010; Barling, Heads & Martill, 2013).

The fossil Minutoma yathribi Kaddumi, 2005 is currently the oldest described putative chalcidoid wasp from Jordanian amber, dated about 115 million years old (myo) (Kaddumi, 2005). It was placed in Mymaridae, which is considered to be the sister group to all other chalcidoid families (Heraty et al., 2013). Heraty et al. (2013), however, commented that the photo of M. yathribi rather suggests affiliation with Bouceklytinae, an extinct subfamily of uncertain placement. Kaddumi (2005) also mentioned a putative eupelmid fossil, which was not formally described. The family assignment of the concerned fossil is questionable as the metasomal and wing venational characters depicted in Kaddumi, (2005, figs. 95–97) are characteristic for Scelioninae (Platygastridae) (own observation). Myanmymar aresconoides Poinar & Huber, 2011 represents the oldest verified fossil record of Chalcidoidea, dating back to the Early Upper Cretaceous, approximately 99 mya (Shi et al., 2012). Although there are some reports of Eulophidae and Chalcididae from the transition between the Upper and Lower Cretaceous, no information concerning their validity is available (Penney, 2010).

Schmidt et al. (2010) reported Eulophidae, Trichogrammatidae and Mymaridae from Ethiopian amber, which they dated through chemical and spectroscopic methods to an Upper Cretaceous origin (around 94 mya). Though the family identifications might be right, doubt was raised concerning the age of Ethiopian amber. Coty, Lebon & Nel (2016) described a myrmecine ant from the same deposit, which could readily be described in the tribe Crematogastrini, suggesting through phylogenetic dating that the specimen cannot be of Cretaceous age. Subsequent revised gas chromatography and infrared spectroscopy analyses showed, that indeed, though not completely unequivocal, evidence strongly suggested that Ethiopian amber is of Cenozoic origin, probably at least 50 million years younger than formerly suspected (Coty, Lebon & Nel, 2016). Currently, the oldest verified record of the families Trichogrammatidae and Aphelinidae are from Baltic amber, approximately 44 myo (Burks et al., 2015).

From the Upper Cretaceous Canadian amber (∼75 myo), fossil Tetracampidae and Trichogrammatidae were recorded by Yoshimoto (1975). Of the four genera described by Yoshimoto (1975) within Mymaridae (Carpenteriana, Macalpinia, Protooctonus and Triadomerus), Protooctonus was later transferred to Mymarommatidae and synonymized under Archaeromma Yoshimoto, 1975 (Gibson, Read & Huber, 2007). Enneagmus Yoshimoto, 1975, originally described within Trichogrammatidae, was transferred by Huber (2005) to Mymaridae. The placement of Distylopus, Bouceklytus and Baeomorpha within Tetracampidae by Yoshimoto (1975) was considered erroneous and even the position of Distylopus within Chalcidoidea was presumed unlikely as stated by Gumovsky & Perkovsky (2005) and Heraty & Darling (2009). After a recent revision, Distylopus and Bouceklytus are now regarded as Chalcidoidea incertae sedis and Baeomorpha and its respective subfamily Baeomorphinae were transferred to Rotoitidae (Gumovsky, Perkovsky & Rasnitsyn, 2018). McKellar & Engel (2012) additionally mention Torymidae and Eupelmidae as possibly present in Canadian amber, although the specimens have not been thoroughly studied to date.

A putative member of Pteromalidae, Parviformosus wohlrabeae Barling, Heads & Martill, 2013, was described from limestone originating from the Crato formation, dated to the Aptian period, about 110 mya. Because of its age, it might be considered as one of the oldest known fossils of Chalcidoidea, but evidence for its placement is ambiguous because none of the diagnostic features of Chalcidoidea was preserved (Barling, Heads & Martill, 2013; Farache et al., 2016). It was placed within Pteromalidae only because of a putative habitus resemblance to Sycophaginae (now Agaonidae sensu Heraty et al., 2013). The limited morphological characters of P. wohlrabeae need to be reassessed before phylogenetic conclusions can be drawn from this fossil. The original placement of P. wohlrabeae in Pteromalidae is in this case highly problematic, because the family, in its current concept, is indicated to be polyphyletic (Campbell et al., 2000; Krogmann & Vilhelmsen, 2006; Heraty et al., 2013).

We here contribute to the scarce Cretaceous fossil record of Chalcidoidea by describing three new fossil genera and species within a new family. These fossils lack synapomorphies with any of the currently described chalcidoid families, but possess many putatively plesiomorphic features, suggesting a basal position within Chalcidoidea.

Material & Methods

Specimens

Four specimens in four different pieces of Burmese amber were examined. Burmese amber is of Upper Cretaceous origin, approximately 99 my old (Shi et al., 2012). Additional information about the geographical origin of the individual pieces is not known. All pieces are deposited in the amber collection of the State Museum of Natural History, Stuttgart, Germany (SMNS).

Imaging

Imaging was done, using a MZ 16 APO Leica® microscope, with an attached DXM 1200 Leica® camera. The images were generated by stacking single images using the Automontage® technique and the program Helicon Focus Pro® (Vers. 6.7.1; Helicon Soft, Kharkov, Ukraine). For additional and detail imaging as well as measurements the digital microscopes Keyence VHX 600 and VHX 5000 were used. Adobe Photoshop® CS4 Version: 11.0.2 was used to process all images. Drawings were made, using a camera lucida on a Leica® M205 C microscope. Digitalization of the drawings and arrangement of the image plates was done with Adobe Illustrator® CS4 (Version: 14.0.0).

Terminology

Terminology follows the Hymenoptera Anatomy Ontology (HAO) (Yoder et al., 2010). Abbreviations listed in Table 1 are used throughout the text and illustrations.

Table 1 Abbreviations for morphological structures.

HAO-Numbers provide direct links to referenced structures in the HAO database (http://glossary.hymao.org).

Abbreviation	Morphological structure	HAO-Number	
ax	Axilla	HAO_0000155	
bv	Basal vein	HAO_0000170	
cer	Cercus	HAO_0000191	
cx1	Procoxa	HAO_0001122	
cx2	Mesocoxa	HAO_0000635	
cx3	Metacoxa	HAO_0000587	
F1-12	Flagellomeres 1-12	HAO_0000342	
frn	Frenum	HAO_0000355	
lbr	Labrum	HAO_0000456	
mps	Multiporous plate sensillum	HAO_0000640	
msc	Mesoscutum	HAO_0000575	
Mt	Metasomal tergite	HAO_0002005	
Mt8+9	Syntergum	HAO_0000987	
no1	Pronotum	HAO_0000853	
no3	Metanotum	HAO_0000603	
not	Notaulus	HAO_0000647	
ov	Ovipositor	HAO_0000679	
pl1	Propleuron	HAO_0000862	
pl2	Mesopleuron	HAO_0000566	
pre	Prepectus	HAO_0000811	
prp	Propodeum	HAO_0000051	
ptl	Petiole	HAO_0000020	
set	Seta	HAO_0002299	
sctl	Mesoscutellum	HAO_0000574	
tfs	Transfacial sulcus	HAO_0002016	
tgl	Tegula	HAO_0000993	
tps	Tentorial pit	HAO_0000999	

Cladistic analysis

Morphological cladistic analyses were performed using the 233 characters from Heraty et al. (2013). Their comprehensive matrix, encompassing 19 families, 78 subfamilies, 268 genera and 283 species of Chalcidoidea was used as basis for the here conducted phylogenetic analysis. Due to preservation and inaccessibility, some characters could not be scored for the fossils without reasonable doubt and were marked as unknown “?” (Appendix S1). Analyses were conducted using the program TNT ver. 1.5 (Goloboff, Farris & Nixon, 2008) following Heraty et al. (2013) in analysis setup. A sectorial search, with equally weighted characters, under New Technology methods was performed, using a ratchet weighting probability of 5% with 50 iterations, tree-drifting of 50 cycles, tree-fusing of five rounds and a best score hit of 10 times. New Technology searches in TNT provide refined algorithms more effective than simple branch swapping techniques applied in traditional searches, leading to shorter analyze times, especially in large datasets (Goloboff, Farris & Nixon, 2008). Nevertheless, traditional searches with and without implied weighting were conducted as well to test consistency of the results. Dependent on the used concavity constant (k), implied weighting aims to decrease the phylogenetic impact of supposed homoplasious characters, in comparison to equal weighting, (Congreve & Lamsdell, 2016). Morphological datasets of Chalcidoidea are reported to include a multitude of potentially homoplasious characters (Krogmann & Vilhelmsen, 2006; Heraty et al., 2013), therefore implied weights of k = 1, 3, 5, 10, 15, 20, 25, 30, 35, 40, 45, 50, 55, and 60 were used with 1,000 replications and Tree Bisection and Reconnection (TBR) for the analyses.

Nomenclature

The electronic version of this article in Portable Document Format (PDF) will represent a published work according to the International Commission on Zoological Nomenclature (ICZN), and hence the new names contained in the electronic version are effectively published under that Code from the electronic edition alone. This published work and the nomenclatural acts it contains have been registered in ZooBank, the online registration system for the ICZN. The ZooBank LSIDs (Life Science Identifiers) can be resolved and the associated information viewed through any standard web browser by appending the LSID to the prefix http://zoobank.org/. The LSID for this publication is: LSID urn:lsid:zoobank.org:pub:B936D52D-7165-47CE-9C3E-0B79A17AC5AC. The online version of this work is archived and available from the following digital repositories: PeerJ, PubMed Central and CLOCKSS.

Results

Systematic Paleontology

Diversinitidae fam. n.	
LSID urn:lsid:zoobank.org:act:017E601E-FB88-4821-8EA7-16228EC61C37	

Type genus. Diversinitus gen. n.

Diagnosis. Antenna 13-segmented in male (Figs. 1B–1D, 2A, 3A and 3B) and female (Figs. 4C, 4E and 5A), with eight funiculars and 3-segmented clava, including a distinct terminal button; all funiculars (including F1) with multiporous plate sensilla (Figs. 1C, 4C and 4E). Eyes large, without pilosity, inner margins not divergent ventrally (Figs. 1B and 4B). Occipital carina absent. Labrum exposed below clypeus (Figs. 1B and 4B), semicircular, flap-like with setae at least at apical margin, broadly contiguous with clypeal margin. Mandibles two toothed (Figs. 2A and 3A). Pronotum lacking collar (Figs. 3A, 5A and 5B). Mesosoma with independent, large and triangular, laterally exposed prepectus (Figs. 3A, 5A and 5B). Mesothoracic spiracle situated at lateral margin of mesoscutum, at juncture of pronotum and prepectus. Mesonotum with notauli deep and complete (Figs. 1D, 3B, 5A and 5B). Mesoscutellum with frenum distinguishable (Figs. 1D, 3B and 4F). Mesopleuron concave with acropleuron not enlarged. Fore wing with short marginal fringe. Basal vein at least apically pigmented (Figs. 1E and 4A). Uncus elongate, bent in direction of postmarginal vein (Figs. 1E, 4A and 4D). Postmarginal vein distinctly longer than marginal vein (Figs. 1E, 4A and 4D). Hind wing normal, membrane extending to base of wing, three hamuli, first straight (Figs. 1E and 4A). Tibial spur formula 1:1:2; protibial spur slightly curved, slender, simple tip; mesotibial spur slender and straight. Tarsus on all legs five segmented. Metasoma with Mt8 and Mt9 fused into syntergum (Mt8+9) in both sexes (Figs. 3A, 3B, 5A and 5B). Cercus peg-like (Figs. 3A, 5A and 5B).

Figure 1 Digital microscopic images of Diversinitus attenboroughi holotype, male.

(A) Dorsal habitus. (B) Head frontal. (C) Antenna detail dorsal. (D) Head and mesosoma dorsal. (E) Wings left side. Scale bars: (A, C) 0.5 mm, (B, E) 2.5 mm, (D) 0.2 mm. Abbreviations: ax, axilla; F1/11, funicular 1/11; frn, frenum; lbr, labrum; mps, multiporous plate sensilla; msc, mesoscutum; no1, pronotum; not, notaulus; sctl, scutellum; ptl, petiole. Photos by M Haas.

Figure 2 Digital microscopic images of Diversinitus attenboroughi lateral habitus, males.

(A) Holotype. (B) Paratype. Scale bars: 0.5 mm. Photos by M Haas.

Figure 3 Habitus drawings of Diversinitus attenboroughi holotype, male.

(A) Habitus lateral. (B) Habitus dorsal. Scale bars: 0.5 mm. Abbreviations: ax, axilla; bv, basal vein; cer, cercus; cx1∕2∕3, pro-/meso-/metacoxa; F1/11, funicular 1/11; frn, frenum; msc, mesoscutum; Mt2, metasomal tergum 2; Mt8+9, syntergum; no1, pronotum; not, notaulus; pl1, propleuron; pre, prepectus; prp, propodeum; ptl, petiole; sctl, scutellum; tgl, tegula. Drawings by M Haas.

Figure 4 Digital microscopic images of Burminata caputaeria and Glabiala barbata, female.

(A, B, C) Burminata caputaeria (A) lateral habitus. (B) Head frontal. (C) Right antenna lateral. (D, E, F, G) Glabiala barbata (D) lateral habitus. (E) Left antenna dorsal. (F) Mesosoma lateral. (G) Mesosoma dorsal. Scale bars: (A, D) 0.5 mm, (B, C, E, F, G) 0.1 mm. Abbreviations: ax, axilla; F1/11, funicular 1/11; frn, frenum; lbr, labrum; mps, multiporous plate sensilla; msc, mesoscutum; no1, pronotum; no3, metanotum; sctl, scutellum; tfs, transfacial sulcus; tps, tentorial pits. Photos by M Haas.

Figure 5 Habitus drawings of female holotypes of Burminata caputaeria (A) and Glabiala barbata (B).

Scale bars: 0.5 mm. Abbreviations: ax, axilla; bv, basal vein; cer, cercus; cx1∕2∕3, pro-/meso-/ metacoxa; F1/11, funicular 1/11; msc, mesoscutum; Mt2, metasomal tergum 2; Mt8+9, syntergum; no1, pronotum; not, notaulus; ov, ovipositor; pl1∕2, pro-/mesopleuron; pre, prepectus; prp, propodeum; sctl, scutellum; tgl, tegula. Drawings by M Haas.

Key to species of Diversinitidae

1.	Head distinctly towering over mesosoma (Fig. 4A). Pronotum almost as long as mesoscutum (Figs. 4A and 5A). Basal cell bare, basal vein nearly completely pigmented (Figs. 2A, 4A). Ovipositor protruding about half the length of gaster (Fig. 4A). [only female known]	Burminata caputaeria	
-	Head only slightly towering over mesosoma (Figs. 2A, 2B and 4D). Pronotum short, about 1/4 length of mesoscutum (Figs. 1A and 4G) Basal cell pilose (Fig. 1E), basal vein only apically pigmented (Fig. 2B). Ovipositor only slightly protruding gaster (Fig. 4D).	2.	
2.	Gaster sessile (Fig. 4D). Mouthmargin surrounded by long setae (Figs. 4D and 5B). Antenna inserted at about center of face, with toruli closer to each other than to margin of eye. Axilla advanced almost 1/4 length of mesoscutum (Fig. 4G). Speculum on forewing present (Fig. 4D). Protibia with a row of stout setae on anterior margin. [only female known]	Glabiala barbata	
-	Gaster distinctly petiolate (Figs. 1A and 1D). Mouthmargin not surrounded by long setae (Fig. 1B). Antenna inserted in lower third of face, with toruli closer to eye margin than to each other (Fig. 1B). Axilla not advanced (Fig. 1D). Speculum on forewing absent (Fig. 1E). Protibia without row of stout setae on anterior margin. [only males known]	Diversinitus attenboroughi	

Diversinitus gen. n.	
LSID urn:lsid:zoobank.org:act:F8B422B0-C83B-4718-8042-D7F07EA0DF7F	

Type species. Diversinitus attenboroughi sp. n.

Diagnosis. Antenna inserted in lower third of face (Fig. 1B). Clypeus transverse. Scape ventrally expanded (Figs. 2A and 3A). Pronotum less than 1/4 length of mesoscutum (Figs. 1D and 3B). Axilla not advanced. Frenum anteriorly delimited by deep frenal groove (Figs. 1D and 3A). Fore wing completely pilose, i.e., speculum absent (Fig. 1E), basal vein only anteriorly pigmented. Gaster distinctly petiolate (Fig. 1D).

Etymology. The generic name Diversinitus is composed of two parts. The first being “Divers-”, originating from the Latin adjective “diversus”, meaning diverse or different. The second part, “-initus”, is the Latin noun “initus” translating to “origin” or “start”. Together the two parts can be translated to “origin of diversity”, referring to the age of the fossil and the diversity which evolved since its appearance in the Upper Cretaceous. The generic name is masculine in gender.

Diversinitus attenboroughi sp. n. (Figs. 1–3)	
LSID urn:lsid:zoobank.org:act:3840E4D4-46A6-4192-8052-20E561DD913F	

Diagnosis. As for the genus.

Male. Measurements: (h), holotype; (p), paratype. Total body length, excluding protruded aedeagus 1.67 mm (h), metasoma of paratype destroyed.

Head. In frontal view oval, bare, broader than rest of body, breadth 0.41 (p)–0.52 mm (h), height 0.34 mm (h), length in dorsal view 0.23 (p)–0.29 mm (h). Foramen magnum situated higher than half height of head. Eye length 0.25 mm (h), height 0.28 mm (h), distance between eyes 0.21 (p)–0.23 mm (h). Transfacial sulcus indiscernible. Antennal scrobes probably shallow. Clypeus transverse, apically truncate, tentorial pits absent, dorsal margin straight. Mandible at least two times as long as broad with slight curvature and few short setae on outer surface. Maxillary palps with at least three segments. Labial palps with at least two segments. Malar space shorter than 1/3 length of an eye.

Antenna. Inserted in lower third of face, hardly above ventral level of eyes, with toruli closer to edge of eyes than to each other. Scape ventrally expanded and broadened over most of its length, not reaching median ocellus. Pedicel length, measured laterally, 0.05 (h)–0.06 mm (p) and breadth 0.04 mm (h, p). F1 subquadrate, fully developed (not anelliform); following funiculars increasingly more transverse and broadening distally, F1–F4 with sides diverging (subconical), F5–F8 more parallel sided (cylindrical) and asymmetrically formed, connections between segments rather slanted; F1 dorsolateral length (mm): width (mm) = 0.03(p)–0.04 (h): 0.03 (p)–0.04 (h), F2 = 0.02 (h, p): 0.04 (h, p), F3 = 0.02 (p)–0.03 (h): 0.04 (h, p), F4 = 0.2 (p)–0.03 (h): 0.04 (h, p), F5 = 0.03 (h, p): 0.04 (h, p), F6 = 0.03 (h, p): 0.04 (h, p), F7 = 0.03 (p, h): 0.04 (p)–0.05 (h), F8 = 0.03 (p)–0.04 (h): 0.04 (p)–0.05 (h). Clava differentiated, sutures rather straight; F9 length (mm): width (mm) = 0.03 (h, p): 0.04 (p)–0.05 (h), F10 = 0.02 (h, p): 0.03 (p)–0.04 (h), F11 + F12 = 0.02 (p)–0.04 (h): 0.02 (p)–0.03 (h).

Mesosoma. Length 0.60 (p)–0.74 mm (h), arched. Pronotum bare, posteriorly deeply emarginated, u-shaped, medially much shorter (0.06 (p)–0.07 mm (h)) than mesoscutum (0.25 (p)–0.27 mm (h)), regularly reticulate. Prepectus slightly convex, lightly sculptured, with thin shiny rim along its dorsal and posterior margin. Tegula much smaller than prepectus. Mesonotum finely regularly reticulate and with very sparse, short pilosity. Mesoscutum slightly shorter than wide; notauli reaching transscutal articulation, widely separated posteriorly. Mesoscutellum length 0.23 (p)–0.26 mm (h), with frenum delimited anteriorly by deep frenal groove (length: 0.05 (p)–0.06 mm (h)); axillae not advanced, widely separated at transscutal articulation. Metapleuron small, bare. Metanotum length 0.06 (p)–0.07 mm (h), with smooth metascutellum not reaching anterior margin of metanotum, lateral panel foveolate. Propodeum transverse, rectangular, slightly arched, length 0.09 (p)–0.11 mm (h), with coarse irregular sculpture, lateral propodeal callus bare; spiracles round to slightly elliptical.

Wings. Fore wing hyaline, immaculate, entirely pilose; humeral plate with at least three setae; basal vein apically pigmented and angled relative to submarginal vein at about 10–15°; marginal vein slightly thickened relative to postmarginal vein; stigmal vein about 0.5 times length of marginal vein; uncus bent at angle of about 95–100° in direction of postmarginal vein, almost reaching it; postmarginal vein not reaching apex of wing, 1.5 times as long as marginal vein. Hind wing apical 2/3 pilose, rest relatively bare; posterior marginal fringe moderately long.

Legs. Pro- and metacoxa larger than mesocoxa; metacoxa dorsally bare, except few hairs posteriorly. Protibial setae inconspicuous and short. Basitarsal comb not visible. Metatibia laterally flattened, bearing two spurs, one robust, the other short and more slender.

Metasoma. Petiole (Mt1) cylindrical distinct and reticulate, length 0.09 mm (h), breadth 0.06 mm (h). Gaster of holotype 0.66 mm in length, lanceolate; terga smooth and bare except of Mt6–Mt8+9 with longitudinal rugosity and lateral setae, hindmargins straight, length of terga of holotype: Mt2: 0.24 mm, Mt3: 0.07 mm, Mt4: 0.07 mm, Mt5: 0.07 mm, Mt6: 0.11 mm, Mt7: 0.06 mm, Mt8+9: 0.04 mm. Cerci peg-like with few long setae.

Female. Unknown.

Specimen examined. Male holo- (SMNS Bu-4) and paratype (SMNS Bu-5) deposited in the SMNS. The amber piece hosting the holotype also includes syninclusions: three complete Diptera and three further Diptera, which are preserved only in part. Additionally, a Serphitidae (Hymenoptera) is included in the same piece. The amber piece including the paratype also hosts a Platygastridae: Scelioninae (Hymenoptera).

Etymology. Named after the well renowned British broadcaster and naturalist Sir David Frederick Attenborough for his inspiring enthusiasm and devotion to natural sciences. This species was dedicated to Sir Attenborough during his visit to the SMNS on the occasion of his 91st birthday.

Burminata gen. nov.	
LSID urn:lsid:zoobank.org:act:71D5E586-8406-486A-85AC-FA5CA1F293D8	

Type species. Burminata caputaeria sp. n.

Diagnosis. Foramen magnum situated at lower third of head (Fig. 4A). Tentorial pits deep (Fig. 4B). Clypeus transverse (Fig. 4B). Pronotum only slightly shorter than mesoscutum (Fig. 5A). Axilla slightly advanced (Fig. 5A). Fore wing with speculum; basal cell bare; basal vein almost completely pigmented (Fig. 4A). Posterior fringe on hind wing long (Fig. 4A). Ovipositor protruding about half length of gaster (Fig. 4A).

Etymology. The generic name is composed of two parts. The first part “Burmi-”, references the origin of the amber piece whereas the second part, “–nata”, originates from the Latin adjective “natus” translating to “born”. The generic name is feminine in gender.

Burminata caputaeria sp. n. (Figs. 4A–4C and Fig. 5A)	
LSID urn:lsid:zoobank.org:act:AA5C051D-90AB-4D21-80F1-90AE82A8125A	

Diagnosis. As for the genus.

Female. Total body length, excluding protruding ovipositor 1.23 mm.

Head. In frontal view oval, bare, much broader than rest of body, breadth 0.40 mm, height 0.23 mm, dorsal length not measurable. Foramen magnum situated at lower third of head. Eye length 0.17 mm, height 0.16 mm, distance between eyes 0.22 mm. Putative transfacial sulcus anterior to antennal scrobes length 0.10 mm. Antennal scrobes absent. Clypeus transverse, apically truncate, laterally delimited by large tentorial pits, dorsal margin straight. Mandible about 1.6 times as long as broad, rather straight, setae not distinguishable. Maxillary palps with at least three segments. Labial palp segments not countable. Malar space more than 1/3 length of an eye.

Antenna. Inserted at about center of face, at half height of eyes, with toruli slightly closer to edge of eyes than to each other. Scape slender, not flattened, not reaching median ocellus. Pedicel length, measured laterally, 0.04 mm and breadth 0.04 mm. F1 subquadrate, subconical, fully developed (not anelliform); following funiculars transverse to quadrate, conical, connections between segments rather slanted; F1 lateral length (mm): width (mm) = 0.03: 0.04, F2 = 0.03: 0.04, F3 = 0.03: 0.04, F4 = 0.04: 0.04, F5 = 0.04: 0.04, F6 = 0.04: 0.04, F7 = 0.04: 0.04, F8 = 0.04: 0.04. Clava differentiated, sutures oblique, F9 length (mm): width (mm) = 0.04: 0.04, F10 = 0.03: 0.04, F11 + F12 = 0.03: 0.03.

Mesosoma. Length 0.49 mm, weakly arched. Pronotum bare except of few long setae on hind margin, posteriorly slightly emarginated, medially only slightly shorter (0.12 mm) than mesoscutum (0.15 mm), regularly finely reticulate. Prepectus convex, lightly sculptured, with thin shiny rim along its dorsal and posterior margin. Tegula much smaller than prepectus. Mesonotum, finely regularly reticulate, largely bare, only few single setae on lateral lobe of mesoscutum, msoscutellum and axilla. Mesoscutum breadth not measurable; notauli reaching transscutal articulation, widely separated posteriorly. Mesoscutellum length 0.14 mm, with frenum short (0.02 mm), delimited anteriorly by shallow frenal groove; axillae slightly advanced, widely separated at transscutal articulation. Metapleuron small and triangular, bare. Metanotum and propodeum hardly discernable because of cracked amber and air inclusions, propodeum apparently arched.

Wings. Fore wing hyaline, immaculate, speculum present, basal cell bare, costal cell pilose throughout; humeral plate with at least two setae; basal vein almost completely pigmented, angled relative to submarginal vein at about 27°; marginal vein as thick as postmarginal vein; stigmal vein about 0.4 times length of marginal vein; uncus bent at angle of about 110° in direction of postmarginal vein, almost reaching it; postmarginal vein not reaching apex of wing, 1.6 times as long as marginal vein. Hind wing apical 2/3 pilose, rest relatively bare; posterior marginal fringe long.

Legs. Pro- and mesocoxa about same size, metacoxa slightly larger, dorsally completely bare. Protibial setae inconspicuous and short. Basitarsal comb not visible. Metatibia hardly flattened, bearing two equally short and robust spurs.

Metasoma. Petiole (Mt1) indistinct. Gaster lanceolate, length excluding ovipositor 0.52 mm; terga smooth and bare except dorsal surface of Mt7 and Mt8+9 with longitudinal rugosity, hindmargins straight, length of terga: Mt2: 0.12 mm, Mt3: 0.04 mm, Mt4: 0.04 mm, Mt5: 0.05 mm, Mt6: 0.06 mm, Mt7: 0.11 mm, Mt 8+9: 0.09 mm. Cercus peg-like, appearing to be slightly spatulate, arising from under syntergum, bearing at least three setae. Hypopygium folded downwards, slightly longer than half of gaster. Ovipositor protruding about half length of gaster, third valvulae broad.

Male. Unknown.

Specimen examined. The holotype (SMNS Bu-304) is deposited in the SMNS. Besides the holotype the amber piece also includes two Diptera and one Platygastridae: Scelioninae (Hymenoptera), amongst parts of other insects.

Etymology. The specific epithet “caputaeria” consists of two parts originating from the Latin noun for “head” (caput) and the adjective for “towering up” (aerius), referring to the lowly situated foramen magnum, leaving the head protruding especially high over the pronotum. The species name is treated as an adjective.

Glabiala gen. nov.	
LSID urn:lsid:zoobank.org:act:10644623-4534-4848-B961-1E608CBB773B	

Type species. Glabiala barbata sp. n.

Diagnosis. Head densely pilose, with mouth margin surrounded by especially long setae (Figs. 4D and 5B). Clypeus quadrate. Toruli situated at about center of face, closer to each other than to margin of eyes. All funiculars rather thistle shaped (Fig. 4E). Pronotum and mesonotum with dense, short pilosity (Fig. 4F). Pronotum about 1/3 the length of the mesoscutum (Figs. 4G and 5B). Axillae advanced about 1/4 the length of the mesoscutum (Fig. 4G). Frenum large, delimited by deep frenal groove (Fig. 4F). Lateral propodeal callus with dense pilosity. Fore wing with speculum (Fig. 4D); basal cell pilose, basal vein only anteriorly pigmented. Metacoxa dorsally with short pilosity. Ovipositor hardly protruding apex of gaster (Fig. 4D).

Etymology. The name consists of two parts originating from the Latin words for “hairless” (glabellus) and “wing” (ala), referring to the distinct speculum on the wing of the specimen. The generic name is feminine in gender.

Glabiala barbata sp. n. (Figs. 4D–4G and Fig. 5B)	
LSID urn:lsid:zoobank.org:act:01C89C3D-E207-4544-A5AD-3BA80EFE61CB	

Diagnosis. As for the genus.

Female. Total body length, excluding protruding ovipositor: 2.21 mm.

Head. Frontal view largely blocked, appearing trapezoid, finely pilose, except quite long pilosity on gena and mouth margin, about as broad as body, actual breadth and height not measurable. Foramen magnum situated higher than half height of head. Eye length 0.23 mm, height 0.27 mm, distance between eyes not measurable. Transfacial sulcus not discernable. Antennal scrobes absent. Clypeus quadrate with subparallel sides, apically truncate, tentorial pits absent, dorsal margin straight. Mandible not measurable, appearing broad and straight, with numerous longer setae on its outer surface. Maxillary palps probably with four segments. Labial palps with at least two segments. Malar space about 1/3 length of an eye.

Antenna. Inserted at about center of face (direct frontal view blocked), slightly below half height of eyes, with toruli closer to each other than to eyes. Scape slightly broadened, not reaching median ocellus. Pedicel lateral length not assessable. F1 subconical fully developed (not anelliform), distal funiculars more transverse, F2–F8 appearing thistle-shaped, with F2–F7 asymmetrically shaped, connections between segments rather slanted; F1 lateral length (mm): width (mm) = 0.05: 0.05, F2 = 0.06: 0.05, F3 = 0.05: 0.05, F4 = 0.06: 0.05, F5 = 0.05: 0.05, F6 = 0.05: 0.06, F7 = 0.05: 0.06, F8 = 0.05: 0.06. Clava not clearly differentiated, segments separated by deep, rather straight sutures, F9 length (mm): width (mm) = 0.04: 0.05, F10 = 0.05: 0.05, F11 + F12 = 0.05: 0.04.

Mesosoma. Length 0.96 mm, weakly arched. Pronotum densely shortly pilose, posteriorly deeply emarginated, u-shaped, medially much shorter (0.12 mm) than mesoscutum (0.4 mm), regularly reticulate. Prepectus almost flat, lightly sculptured, view on rim not clear. Tegula smaller than prepectus. Mesonotum regularly reticulate and densely, shortly pilose. Mesoscutum about 2/3 as long as wide; notauli reaching transscutal articulation, widely separated posteriorly. Mesoscutellum length 0.31 mm, with frenum delimited anteriorly by deep frenal groove (length: 0.07 mm); axillae strongly advanced, about 1/4 length of mesoscutum, widely separated at transscutal articulation. Metapleuron small and triangular, with few scattered setae. Metanotum length 0.06 mm, with smooth metascutellum not reaching anterior margin of metanotum, lateral panels prominent, foveolate. Propodeum transverse, rectangular, relatively flat, length 0.11 mm, reticulation regular, lateral propodeal callus with dense and long pilosity; spiracles round to slightly elliptical.

Wings. Fore wing hyaline, immaculate, speculum present, basal cell pilose, costal cell pilose throughout; humeral plate with at least two setae; basal vein apically pigmented and angled relative to submarginal vein at about 9°; marginal vein slightly thickened relative to postmarginal vein; stigmal vein about 0.4 times length of marginal vein; uncus bent at angle of about 95° in direction of postmarginal vein, almost reaching it; postmarginal vein not reaching apex of wing, 1.6 times as long as marginal vein. Hind wing apical 1/2 densely pilose, the rest relatively bare; posterior marginal fringe short.

Legs. Pro-, meso- and metacoxa about same size, metacoxa dorsally with short pilosity. Protibia with stout setae on anterior margin, other setae more inconspicuous. Basitarsal comb longitudinal. Metatibia laterally flattened bearing two slender spurs, subequal in length.

Metasoma. Petiole (Mt1) indistinct. Gaster lanceolate, length excluding ovipositor 0.98 mm; terga smooth and bare, hindmargins straight, length of terga: Mt2: 0.21 mm, Mt3: 0.09 mm, Mt4: 0.15 mm, Mt5: 0.18 mm, Mt6: 0.14 mm, Mt7: 0.11 mm, Mt8+9: 0.1 mm. Cercus peg-like, club-shaped, arising from under syntergum, bearing at least three setae. Hypopygium folded downwards, slightly longer than 2/3 of the gaster. Ovipositor protruding about length of Mt8+9, third valvulae broad.

Male. Unknown

Specimen examined. Female holotype (SMNS Bu-303) deposited in the SMNS. The piece of amber was cut to reveal a better view of the specimen. Both pieces are free of other inclusions.

Etymology. The specific epithet “barbata” is the feminine form of the adjective “barbatus” which means “bearded” and refers to the setose lower face of the specimen. The species name is treated as an adjective.

Taxonomic remarks

It may seem counterintuitive to place the only two known males of Diversinitidae in a separate genus than the two females, especially since sexual dimorphism is widely spread in Chalcidoidea, most notably in Agaonidae and Eupelmidae resulting in a separation of sexes in morphological analysis of females and males, when coded separately (Krogmann & Vilhelmsen, 2006; Heraty et al., 2013). In most other chalcidoids however, those modifications do not include severe changes to the body plan and are often confined to body size (Hurlbutt, 1987) and antennal characters (Barlin & Vinson, 1981). Males of D. attenboroughi differ from both known females of Diversinitidae by the absence of a speculum on the forewing (versus presence of speculum), an elongate petiole (versus a transverse petiole) and an antennal insertion in the lower 1/3 of the face (versus an insertion near center of face). In addition, they also lack each of the diagnostic characters of the other two females (see below) so that a separate generic placement seems to be justified.

Furthermore, we consider the two females as not congeneric based on significant morphological differences: Glabiala barbata differs from B. caputaeria in having the foramen magnum situated higher than half the height of the head (versus lower third of head), a pronotum only 1/3 length of mesoscutum (versus slightly shorter than mesoscutum), distinctly advanced axillae (versus slightly advanced), a large and clearly anteriorly delimited frenum (versus short and shallowly delimited) and a pilose basal cell on the forewing (versus a bare basal cell).

Results of cladistics analyses

The new technology analysis in TNT found 39 most parsimonious trees (5,395 steps) with the strict consensus tree being 5,861 steps long. The general topology of Heraty et al. (2013) could largely be retrieved (Fig. 6). As in Heraty et al. (2013) the following families appeared as monophyletic: Agaonidae, Chalcididae, Encyrtidae, Eurytomidae, Leucospidae, Mymaridae, Rotoitidae, Signiphoridae, Torymidae (including Megastigminae) and Trichogrammatidae. Contrary to Heraty et al. (2013), Aphelinidae and Eucharitidae could be retrieved as monophyletic as well. In the unweighted new technology analysis Mymarommatoidea was nested within Chalcidoidea as part of a larger clade containing the chalcidoid families Aphelinidae, Mymaridae, Rotoitidae and Signiphoridae, as well as few members of Tetracampidae and Eulophidae. Leucospidae were recovered as sistergroup to all other Chalcidoidea, including Mymarommatoidea. The fossils were recovered as a monophyletic group with Micradelus rotundus Walker, 1834 as sister taxon, nested within a large polytomy. Monophyly of the fossils could be retrieved in all analyses, however general tree topology changed considerably between different analyses. Using a traditional search without implied weighting (Appendix S2), Diversinitidae were recovered as sistergroup of all other Chalcidoidea with the inclusion of Mymarommatoidea. Mymaridae as well as Rotoitidae clustered in deeper clades far from the base of the tree. Using a traditional search with implied weights (Appendix S2), Mymarommatoidea were almost always recovered as sistergroup of Chalcidoidea (except k = 45), but topology changed drastically with increasing k value, as did the position of the fossils within the tree. In most analyses with k values below 30, the fossils were closely affiliated with the pteromalid genera Habritys brevicornis (Ratzeburg, 1844), Cheiropachus quadrum (Fabricius, 1787) and other interchanging groups. Above a k of 30, M. rotundus was recovered as a sistertaxon (k = 35 and 55) or only Cheiropachus quadrum (k = 40), Diversinitidae were sister to all Chalcidoidea including Mymarommatoidea (k = 45) or they were recovered close to Platynocheilus cuprifrons (Nees, 1834) and some Ormocerinae (k = 50 and 60).

Figure 6 Phylogenetic placement of Diversinitidae within Chalcidoidea based on morphological characters.

Strict consensus tree calculated from 39 trees (tree length = 5861, CI = 0.077, RI = 0.567, 232 characters and 304 taxa, equal weights, new technology search). Yellow box highlights described fossils. Mymarommatoidea, potential sistergroup to all Chalcidoidea, collapsed and highlighted in blue. Green names indicate monophyletic and therefore collapsed families. Red names indicate monophyletic and therefore collapsed pteromalid subfamilies. Grey names indicate single taxa. For more information on the dataset of extant taxa refer to Heraty et al. (2013).

Discussion

The placement of Diversinitidae within Chalcidoidea is well supported by several morphological synapomorphies. One of the key autapomorphies of Chalcidoidea are the structurally unique multiporous plate sensilla (mps) on the antennal funicle, with their apices free of their surrounding antennal cuticle, the lack of an encircling groove around the sensillum and elevation of the multiporous plate above the antennal cuticular level (Barlin & Vinson, 1981; Gibson, 1986; Basibuyuk & Quicke, 1999). Evidently, Diversinitidae have modified sensilla (Figs. 1C, 4C and 4E), which are raised above the antennal surface and have their apices not completely surrounded by the antennal cuticle. Some mps, although not all, even protrude slightly over the funicular apices, as seen with backlighting under high magnification. The lack of an encircling groove cannot be unequivocally confirmed, but overall resemblance to mps of other Chalcidoidea is apparent. Within those groups of Proctotrupomorpha that are most closely related to Chalcidoidea (Peters et al., 2017), few possess mps on their antennae. Only Cynipoidea and the family Pelecinidae within Proctotrupoidea share this feature, but show a quite different sensillar morphology with their sensillae usually only slightly raised above the antennal surface and possessing a groove surrounding the multiporous plate (Basibuyuk & Quicke, 1999). Other Proctotrupoidea, Ceraphronoidea, Platygastroidea and Diaprioidea possess setiform multiporous sensilla sharing little resemblance with the morphology of chalcidoid mps (Gibson, 1986; Basibuyuk & Quicke, 1999). Even Mymarommatidae, the putative sister group of Chalcidoidea, lack mps (Gibson, 1986; Munro et al., 2011; Heraty et al., 2013).

Another diagnostic feature of Chalcidoidea is the presence of a free, externally visible prepectus between the pronotum and mesopleuron, which separates the pronotum from the tegula (Gibson, 1985; Gibson, 1999; Gibson, Heraty & Woolley, 1999). Diversinitidae have a large triangular prepectus, neither fused to the pronotum or mesopleuron nor hidden beneath its lateral margin (Figs. 3A, 4A, 5A and 5B). Additionally, like in other chalcidoids, the mesothoracic spiracle is situated between the lateral margin of the mesoscutum and the pronotum directly adjacent to the anterodorsal edge of the prepectus, another autapomorphy of Chalcidoidea that is correlated with its external prepectus. Gibson (1999) hypothesized the more dorsal position of the spiracle compared to other hymenopterans as a derived state. Other hymenopterans having a concealed prepectus or a prepectus that is fused either to the pronotum or mesopleuron have the spiracle originating somewhat more ventrally below the level of the mesoscutum between the pronotum and mesepisternum. In Rotoitidae and Mymaridae, the spiracle is situated between the lateral margin of the mesoscutum and the pronotum, but in Rotoitidae and some Mymaridae the prepectus is slender and more or less concealed under the pronotum. Mymaridae and Rotoitidae are hypothesized as basalmost clades within Chalcidoidea (Gibson, 1986; Munro et al., 2011; Heraty et al., 2013; Peters et al., 2018) and their prepectal structure may represent a transitional state (Gibson, 1999).

Assignment of the fossils to extant chalcidoid families is not possible due to the lack of synapomorphies. The most prominent characteristic of Diversinitidae separating them from all other chalcidoid families, except for some Mymaridae, is the possession of mps on the first flagellomere (F1) in both sexes. Mps on F1 is found in Chalcidoidea only in very few cases. In Mymaridae, most males possess mps on their first flagellomere and also females of very few species (e.g., within the genera Eustochomorpha Girault 1915 and Yoshimotoana Huber, 2015) have them (Heraty et al., 2013; Huber, 2015; Huber, 2017). Some Aphelininae (Aphelinidae) and Eucharitidae also seemingly possess mps on their apparent F1, but this is only because the first two flagellomeres are fused (Heraty et al., 2013). In Diversinitidae, the first visible flagellomere is undoubtedly F1 in both sexes. A well-developed F1 that has mps is hypothesized as plesiomorphic for Chalcidoidea (Heraty et al., 2013), suggesting a basal position of Diversinitidae within Chalcidoidea. During the evolution of Chalcidoidea, the first funicular likely secondarily lost mps in association with the segment being reduced in length to a ring-like segment (anellus) as is suggested by some chalcidoids that have additional funiculars reduced to anelli-like segments that lack mps. In those, comparatively few chalcidoids with F1 lacking mps but being reduced in size, F1 is hypothesized to have been secondarily lengthened (see character 11 in Gibson, 2003).

Burminata caputaeria is the only species in Diversinitidae possessing a discernible line above the scrobal area, corresponding in position and size to a transfacial sulcus (Fig. 4B). A transfacial sulcus or transfacial line, situated below the anterior ocellus right above the antennal scrobes, is found in many, mostly soft-bodied families including Aphelinidae, Encyrtidae, Eulophidae, Eupelmidae (only Phenaceupelmus (Gibson, 1995)), Pteromalidae, Tetracampidae and Trichogrammatidae (Gibson, 1986; Gibson, 1995; Burks et al., 2011; Kim & Heraty, 2012; Heraty et al., 2013). This transfacial sulcus is structurally different from the trabeculae of Mymaridae, which are formed by several interconnected lines of cuticular invaginations, separating the vertex as a distinct sclerite from the face and are therefore regarded as autapomorphic for this family (Königsmann, 1978; Schauff, 1984; Gibson, 1986). Rotoitidae as well as Mymarommatidae lack any indication of a transfacial sulcus (Bouček & Noyes, 1987; Gibson & Huber, 2000; Huber et al., 2008), leaving the ground plan of this character for Chalcidoidea uncertain.

The labrum of Diversinitidae can be described as free, semicircular or rectangular, flap-like and broadly continuous with the clypeal margin. Darling (1988) postulated, that the ground plan structure of the labrum for Chalcidoidea is flap-like, with many evenly distributed setae. Darling (1988) referred to the labrum of Chalcididae as “remarkably uniform and (…) similar to that hypothesized as the ground plan for Apocrita”, being heavily sclerotized and contiguous with the margin of the clypeus, bearing long, tapered setae on the entire surface, arising from distinct sockets. In Pteromalidae, the plesiomorphic state of the labrum is found in Cleonyminae, and the labrum is also exposed in Spalangiinae, Asaphinae, Eunotinae and others, which bear in comparison to Cleonyminae setae only along their apical margin (Darling, 1988). Some Mymaridae also possess an exposed labrum (Heraty et al., 2013; Huber, 2013). In Diversinitidae, the setal pattern is difficult to assess due to refractions within the amber in conjunction with the small size of the specimens. Setae are at least situated along the apical margin in Diversinitidae, but whether they are also found on the surface remains uncertain. If so, the labrum might also be putatively plesiomorphic for Diversinitidae.

Diversinitidae possess a bidentate mandible, which is widely distributed in Chalcidoidea, although a three or more dentate mandible appears to be more common (Bouček & Noyes, 1987; Woolley, 1988; Dzhanokmen, 1996; Gibson, Heraty & Woolley, 1999; Gibson & Huber, 2000; Heraty et al., 2013). The plesiomorphic state for this character is not known and has so far not been discussed for Chalcidoidea comprehensively so that the evolutionary patterns are difficult to assess. Putatively basal chalcidoid families already exhibit varied states of mandible dentation, with Rotoitidae having bidentate mandibles, of which Chiloe micropteron (Gibson & Huber, 2000) has the upper tooth finely serrated (Bouček & Noyes, 1987; Gibson & Huber, 2000). Denticulation in Mymaridae varies greatly, with taxa lacking mandibular teeth (Erythmelus rosascostai Ogloblin, 1934) to taxa with many fine denticles (Eubroncus spp.) (Heraty et al., 2013; Jin & Li, 2014). The mymarid genera Triadomerus (Yoshimoto, 1975) (extinct), Macalpinia (Yoshimoto, 1975) (extinct) and Neotriadomerus (Huber, 2017) (extant) are considered to be the most basal taxa in this family (Huber, 2017). In those early groups mandibular dentation is already differing, with bidentate mandibles in Triadomerus and Macalpinia and four uneven teeth in Neotriadomerus, hampering phylogenetic implications. Outgroup comparisons with Mymarommatoidea and other Proctotrupomorpha (sensu Peters et al., 2017) reveal that also in those groups, mandibular dentation is highly variable (Naumann & Masner, 1985), not permitting a stable hypothesis about the groundplan state for Chalcidoidea. However, Diversinitidae as putative basal group within Chalcidoidea might indicate that bidentate mandibles could be plesiomorphic for at least a smaller subset of chalcidoid taxa.

A frenum is found in Diversinitidae, which is likely a plesiomorphic character state for Chalcidoidea (Krogmann & Vilhelmsen, 2006). Presence is observed in many chalcidoid families and in closely related groups, such as Mymarommatidae, Diapriidae and Platygastridae: Scelioninae (Heraty et al., 2013), suggesting that it is probably part of the ground plan structure for a subgroup of Proctotrupomorpha. Frenal morphology is used in species and subfamily distinction of Torymidae and Pteromalidae (Graham, 1969; Graham & Gijswijt, 1998; Gibson, 2003). The morphological variation of the frenum led to frequent discussions about its homology between different taxonomic groups (Grissell, 1995; Gibson, Heraty & Woolley, 1999; Vilhelmsen & Krogmann, 2006).

Diversinitidae possess peg-like cerci, which are more or less spatulate. This character state has been considered as plesiomorphic in contrast to a button-like cercus (Gibson, 2003) or, alternatively, as an apomorphic character state, which has independently evolved in different chalcidoid groups (Grissell, 1995). Grissell (1995) postulated that though peg-like cerci are found in Agaonidae sensu lato, Eulophidae (Entia Hedqvist, 1974), Pteromalidae (Cea Walker, 1837 and Chromeurytoma Cameron, 1912), Torymidae and Megastigmidae, evolution of this character must have been convergent because positioning of the cerci is different in those groups. On the other hand, Gibson (2003) stated that many other groups have peg-like cerci as well, though most often not as prominent as those listed above, and therefore he considered exerted, basally articulated cerci as plesiomorphic relative to more reduced, plate-like cerci. In Heraty et al. (2013) many taxa were also coded as possessing exerted cerci to various degrees, such as Perilampidae (Brachyelatus sp.), Tetracampidae (Platynocheilus sp.), Signiphoridae (Signiphora sp.), Mymaridae (Borneomymar sp.) and Tanaostigmatidae (Protanaostigma sp.). Outgroup comparison for this character in Heraty et al. (2013) is however not conclusive due to sparse taxon sampling. Mymarommatidae (Mymaromella sp.) was coded as not possessing exerted cerci, compared to Scelioninae (Archaeoteleia mellea Masner, 1968), which show slightly exerted cerci and Diapriidae (Belyta sp.) without coding for this character. The wide distribution of peg-like cerci within Chalcidoidea and its appearance in Mymaridae and Diversinitidae supports the hypothesis that they represent the plesiomorphic state over button-like cerci.

Presenting a solid phylogenetic placement of Diversinitidae within Chalcidoidea is not unequivocally possible. All cladistic analyses provided evidence for monophyly of Diversinitidae, but did not resolve further relationships within Chalcidoidea, because placement of the fossils and general tree topology remained highly variable between different analysis. Although Micradelus rotundus was recovered as sister taxon of Diversinitidae in the new technology analysis and few traditional searches with implied weighting, a true relationship is highly doubtful. Micradelus rotundus belongs to the pteromalid subfamily Pireninae. This subfamily is characterized, though not only, by a reduced number of antennal segments and at least one annellus (Bouček, 1988), which is also the most prominent difference to Diversinitidae, sharing little resemblance to M. rotundus aside from morphologically variable characters like a bidentate mandible, lack of pronotal collar, deep notauli or exposed labrum. Additionally placement of M. rotundus was inconsistent over the different analyses and it behaved like a rogue taxon, jumping between several clades. However, high inconsistencies in the analyses were expected, because the morphology-only analysis in Heraty et al. (2013) was also poorly resolved. Due to the expected high rate of homoplasious characters in morphological datasets of Chalcidoidea (Krogmann & Vilhelmsen, 2006; Heraty et al., 2013), especially the results from analyses with and without implied weighting differed considerably. With increasing k values, the base of the phylogenetic tree was mostly relatively well resolved. Mymmarommatoidea were the sistergroup to Chalcidoidea and Rotoitidae and Mymaridae were retrieved as basal lineages within the superfamily. However, changes in topology of higher relationships were substantial. Through weighing down putative homoplasious characters, implied weighting is capable of better resolving polytomies (Goloboff et al., 2008). This can lead to trees with more correctly resolved clades, but also higher risks of erroneous placements and more inconsistent topologies as demonstrated by Congreve & Lamsdell (2016). Implied weighting can therefore be considered as less conservative over equal weighing of characters. There are conflicting views on whether parsimony analyses (Goloboff, Torres & Arias, 2017), as conducted in this study, or likelihood analyses (O’Reilly et al., 2018) perform better with morphological datasets. A comparison between likelihood and parsimony methods performed by Heraty et al. (2013) on the original dataset, however, resulted in a generally congruent tree with equally poor resolution of taxa. Additionally, probabilistic methods infer an evolutionary model on the data, based on subjective decisions and previous knowledge (Goloboff, Torres & Arias, 2017). We therefore favored the conservative equal weight parsimony analysis over implied weighting and likelihood analyses.

Unfortunately, there is no evident autapomorphic character of Diversinitidae, which would support its monophyly and all characters that exclude this group from existing families are seemingly plesiomorphic (see above). However, based on the unique combination of morphological characters (see diagnosis) and the preliminary results from the cladistic analyses (Fig. 6), we decided to place the new fossils into their own family rather than leaving them unplaced within Chalcidoidea.

Morphologically, Diversinitidae appear to be an early lineage of Chalcidoidea, possessing many putatively plesiomorphic characters (see discussion above). Mymaridae are thought to form the sister group to all remaining Chalcidoidea and can be traced back at least to the mid-Cretaceous (Gibson, 1986; Munro et al., 2011; Heraty et al., 2013). Resemblance between Diversinitidae and Mymaridae is not obvious and they only possess few putatively symplesiomorphic characters, such as an exposed labrum and mps on the true F1 in males and some females. In general, the mymarid body plan is characterized by a number of derived autapomorphies that have not changed much since the Mid Cretaceous (Poinar & Huber, 2011). The phylogenetic position of Diversinitidae can therefore not be established with certainty and several hypotheses are possible. Firstly, Diversinitidae could represent the sister group to all remaining chalcidoids, since they show a multitude of plesiomorphic characters, foremost mps on F1. During chalcidoid evolution mps on F1 might have been lost at first in females (as in most Mymaridae) and subsequently also in males (as in all remaining Chalcidoidea). This would imply, that the prepectus in Diversinitidae was either secondarily enlarged or that Mymaridae and Rotoitidae reduced the prepectal size during their evolution. Diversinitidae might also represent a sistergroup to a smaller subset of Chalcidoidea, suggesting that mps on F1 were independently lost at least twice, once in most females in Mymaridae and once in all other Chalcidoidea. Prepectal size might therefore have been increased in other Chalcidoidea relative to the prepectus in Mymaridae and Rotoitidae.

Biological implications of the new fossils are difficult to draw, because their phylogenetic position is not fully resolved. Egg parasitoidism is hypothesized to be the putative ground plan biology of Chalcidoidea (Heraty et al., 2013; Peters et al., 2018). Diversinitidae share a relatively small body size, which unites nearly all egg parasitizing taxa, but does not necessarily exclude ectoparasitoid groups. Body shape is not indicative, because both ecto- and endoparasitoids can be very diverse in this regard. The length of the ovipositor and its saw-like tip might be indicative for concealed hosts inside plant material.

Conclusion

With the newly described fossils we reduce a significant fossil gap of Chalcidoidea from the Cretaceous. The wasp species described herein provide important new information of chalcidoid evolution because they are early representatives of a parasitoid lineage that was just beginning to evolve. One hundred million years later we merely start to fully appreciate the great morphological diversity and ecological significance of these “green myriads in the peopled grass” (Walker, 1839), which still rank among the least known of all insects. Further Cretaceous fossils will hopefully reduce the fossil gap even further to help us to understand how chalcidoid wasps have evolved and shaped the evolution of their arthropod host groups and associated plant species, as one of the most diverse and influential insect groups that life has ever seen.

Supplemental Information

Table S1 Data matrix constructed for all available members of the family Diversinitidae, using the character list of Heraty et al. (2013)

Click here for additional data file.

Supplemental Information 1 Results of cladistic analyses

Zip file contains results of all performed cladistic analyses sorted by subfolders. Subfolders include final trees, final trees with mapped synapomorphies (.emf format) and a text file (.txt) of the TNT output for more information on analyses setup and tree statistics.

Click here for additional data file.

We thank Patrick Müller (Käshofen, Germany) for the kind donation of the holotype of Burminata caputaeria and Karin Wolf-Schwenninger (SMNS) for providing access to the amber collection and for technical support.

Additional Information and Declarations

Competing Interests

Author Contributions

Data Availability

New Species Registration

The authors declare there are no competing interests.

Michael Haas analyzed the data, prepared figures and/or tables, approved the final draft.

Roger A Burks and Lars Krogmann analyzed the data, approved the final draft.

The following information was supplied regarding data availability:

Coding for cladistic analyses can be extracted from Table S1.

The following information was supplied regarding the registration of a newly described species:

Publication LSID: urn:lsid:zoobank.org:pub:B936D52D-7165-47CE-9C3E- 0B79A17AC5AC;

Diversinitidae:

LSID urn:lsid:zoobank.org:act:017E601E-FB88-4821-8EA7-16228EC61C37;

Diversinitus:

LSID urn:lsid:zoobank.org:act:F8B422B0-C83B-4718-8042-D7F07EA0DF7F;

Diversinitus attenboroughi:

LSID urn:lsid:zoobank.org:act:3840E4D4-46A6-4192-8052-20E561DD913F;

Burminata:

LSID urn:lsid:zoobank.org:act:71D5E586-8406-486A-85AC-FA5CA1F293D8;

Burminata caputaeria:

LSID urn:lsid:zoobank.org:act:AA5C051D-90AB-4D21-80F1-90AE82A8125A;

Glabiala:

LSID urn:lsid:zoobank.org:act:10644623-4534-4848-B961-1E608CBB773B;

Glabiala barbata:

LSID urn:lsid:zoobank.org:act:01C89C3D-E207-4544-A5AD-3BA80EFE61CB.

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
