# Peer review of "A new lineage of Cretaceous jewel wasps (Chalcidoidea: Diversinitidae)"

_PeerJ, doi:10.7717/peerj.4633_

## Round 0.1 · original submission · Minor Revisions

I feel your manuscript is in a good state. I particularly like the meticulous descriptions, the morphological phylogenetic analysis and the extensive discussion. There are however still some crucial points to address before publication:

Taxonomic descriptions: please homogenize your taxonomic descriptions (see comment by reviewer 1).

Justification to erect 3 new genera: Are you sure you can rule out that 2 of the new taxa are not male and females of one and the same taxon (see comments by reviewer 2).

Phylogenetic analyses: I commend the authors for running a cladistic analysis integrating the newly discovered fossils in TNT - something i rarely see and get (even when i asked for it during reviews). Some references stating the general importance of integrating fossils would no be out of place. It would just like the authors to consider to explain better the different between analyses with implied and without implied weights and their potential reasons (implied weights can give inconsistent results: see Congreve and Lamsdell 2016 as well as annotated manuscript). Furthermore, probabilistic methods might surpass parsimony when assessing clade support in morphological datasets (e.g., O´Reilly et al. 2018). I commend you for using the Heraty approach for comparability, but it might still be worth doing or at least discussing this issue.

Support to erect a new family: you erect a new family which is supported by your phylogenetic analyses - however can you rule out these are stem taxa showing up as a monophyletic grouping because of the plesiomorphies (see comment by reviewer 3).

In addition to these points and the points by the reviewers, please address the point listed in the annotated manuscript.

Suggested references:

Congreve, C. R., Lamsdell, J. C. (2016), Implied weighting and its utility in palaeontological datasets: a study using modelled phylogenetic matrices. Palaeontology, 59: 447–462. doi: 10.1111/pala.12236

O'Reilly, J. E., Puttick, M. N., Pisani, D., Donoghue, P. C. J. (2018), Probabilistic methods surpass parsimony when assessing clade support in phylogenetic analyses of discrete morphological data. Palaeontology, 61: 105–118. doi: 10.1111/pala.12330

·

Basic reporting

I read and offered suggested editings throughout the manuscript (e.g. beginning with suggested changes to the title for clarity) except I only read in detail the description of the first newly described genus/species. I did not read the second or third descriptions in detail because as soon as I started the second I realized the descriptions of the three taxa were not rigorously comparable. It is very important that the descriptions be rewritten so that they are rigorously comparable, i.e. that all features described in one description are described in all, and in the same order with the same punctuation/sentence style. This is important because the three genera are being described in a new family and it is important to distinguish between what might be familial, generic or species features. For example, the minimum number of maxillary palps is described for the second genus, but not the first or third. Therefore the reader does not know if the authors just neglected to describe them or they are not visible. It is understood that many features may not be visible in any one fossil, but if visible and important enough to describe in one taxon then the feature should be described in all, even if only as "maxillarly palps not visible". Also, any feature given as a family level feature should not be repeated in the generic description. Finally, the current sentence structure sometimes is very difficult reading because of sometimes very long sentences with major structures separated by semicolons. I suggest, where possible, shorter sentences restricted to single body parts. For the benefit of the reader it is also good practice to always first state what the body part being described is, what view it should be examined from (if necessary), and then the description of the body part.

Experimental design

no comment

Validity of the findings

Within my edited version of the manuscript I suggested one additional feature, the number of mandibular teeth, as a feature I think important to include in the discussion, one instance of two non-congruent features that should be discussed relative to the implications of character evolution and relationships, one instance where I question the authors observation, and a couple of instance I believe require somewhat modified discussion.

Additional comments

My review consists primarily of the editings, suggestions and comments made in balloons to a Word document sent to me at my request from the authors (I refuse to review PDF manuscripts). I will send this directly to the authors and will also attempt to upload as an attachment with my very brief 'formal' review.

Reviewer 2 ·

Basic reporting

Generally good. See marked up text.

Experimental design

Not relevant.

Validity of the findings

Good. But discussion must be modified.

Additional comments

See my attached comments as well as marked up text for suggested changes.

1. English is good and mostly unambiguous. I have edited the descriptions to make them flow better and remove inconsistencies. The literature is relevant except that possibly 3 papers on Mymaridae are missing. I added one and referenced the two others that should at least be consulted for the Discussion section, if not actually referenced in the ms. I don’t know the standards for PeerJ. What is important is whether it conforms to taxonomic standards for Hymenoptera descriptions. It is the editor’s business to ensure that journal standards (format, etc.) are followed. My interest as a review is the content. The figures are relevant but not labelled (no real necessity I think) and described. Raw data is supplied. Expermimenal design, etc. does not apply to this ms, which is taxonomic in nature.
2. Validity of findings. At least two of the taxa are clearly new. I am not sure if the male of one new genus and species is not actually the same as the female of one of the other new genera and species. That possibility was not discussed. It should at least be mentioned and evidence given why it is probably not the opposite sex of one of the two females, e.g. wings with microtrichia on surface vs without?
3. A few facts are simply wrong because the relevant papers on Mymaridae were not consulted (author possibly unaware of them since one only published in 2017). They are drawn to the attention of the authors for serious consideration that should lead to some rewriting of the Discussion.
4. Suggestions on how to improve the ms are given on the Word version, using Track changes.
5. The cladogram presented is well discussed and, as usual for Chalcidoidea, rather ambiguous. Not much can be done about this since higher (family-group) relationships within the superfamily are still very poorly understood, at least beyond Mymaridae and Rotoitidae. The discussion covers the essential points

Annotated reviews are not available for download in order to protect the identity of reviewers who chose to remain anonymous.

Reviewer 3 ·

Basic reporting

no comment

Experimental design

no comment

Validity of the findings

general:
I have a critical comment to the description of the new family Diversinitidae:
As far as I understood, the family is characterized by a combination of plesiomorphic features (which could explain the "monophyly" of the family in the cladistic analysis).
It seems that not a single autapomorphic feature is given for the new family.
Is it not possible that these very old fossils are stem group representatives, that is, very primitive Chalcidoidea?
Please consider: is it really justified to describe a new family without any autapomorphies? I have strong doubts. This could be very easily a paraphyletic taxon or even a polyphyletic one.

Some special comments:

- line 19: instead of "lineages" better "crown groups"?
- line 21: see critical comment on the new description
- line 24: " but confirmed its monophyly" : as I read the manuscript, there are no autapomorphic characters that confirm the monophyly of Diversinitidae, see general comment above
- line 38: submitted, in refs dated as 2017
- line 52, major gap: more explanation would be good
- lines 62 to 64: better place a statement concerning a new systematic placement of certain taxa not in the intro, but better in the discussion
- line 78: can the age of Baltic amber really be dated this precisely? I have doubts.
- line 86+87, erroneous placement: please explain : is this your opinion, or that of other authors?

- line 152 ff: new family:
a differential diagnosis is missing in which the differences to the other families are explained

- line 169, 172: Please explain what means "symmetric" or "asymmetric"
- lines 169 ff: Please label the mentioned characters in the according figures, like axillae in Fig 1D and 2B and 4

- lines 193 and 194: very short diagnosis. Please expand. You could differentiate between apomorphic and plesiomorphic characters and give a differential diagnosis to the other 2 genera

- line 206 ff, description: Please differentiate between the measurements for the holotype and those for the paratype. It should be clear which measurements come from the holotype.

- line 274ff:bettergive a more thorough diagnosis with a differentiation between apomorphic and plesiomorphic characters and add a differential diagnosis to other genera

- line 366: mistake in measurements
- line 388: mistake in measurements

---

## Round 0.2 · Minor Revisions

Thank you for integrating our suggestions. Your paper is as good as accepted, i just found some minor points i would still like your to address before publication. The main point would be to make the analyses with implied weights in some form available as supplementary material. As you did them and discuss them in the text, i would be appropriate to make them available for scientific reproducibility. I only took this opportunity to make some additional suggestions pertaining to language/formatting/structure in the annotated pdf.

---

## Round 0.3 · accepted · Accept

Thank you for implementing my final suggestions. Looking forward to seeing the manuscript published.